# National Trends in Rotavirus Enteritis among Infants in South Korea, 2010–2021: A Nationwide Cohort

**DOI:** 10.3390/children10091436

**Published:** 2023-08-23

**Authors:** Hyun Jee Lee, Yujin Choi, Jaeyu Park, Yong-Sung Choi, Dong Keon Yon, Do Hyun Kim

**Affiliations:** 1Department of Pediatrics, Kyung Hee University Medical Center, Kyung Hee University College of Medicine, Seoul 02447, Republic of Korea; hlee1029@gmail.com (H.J.L.); feelhope@khu.ac.kr (Y.-S.C.); yonkkang@gmail.com (D.K.Y.); 2Department of Korean Medicine, Kyung Hee University College of Korean Medicine, Seoul 02447, Republic of Korea; yuzzin@khu.ac.kr; 3Center for Digital Health, Medical Science Research Institute, Kyung Hee University College of Medicine, Seoul 02447, Republic of Korea; wodb980@naver.com

**Keywords:** COVID-19, rotavirus, vaccination, birth, South Korea

## Abstract

Rotavirus causes a gastrointestinal tract infection that primarily affects young children. Due to the COVID-19 pandemic, individuals infected with the virus were subjected to quarantine measures, with strong emphases on personal hygiene and social distancing. The present study aimed to evaluate the characteristics of rotaviruses and compare the prevalence of rotavirus infection before and during the COVID-19 pandemic. This nationwide representative study was conducted using data acquired from the National Health Insurance Service between 2010 and 2021. We analyzed the data of patients younger than 12 months old who were diagnosed with rotavirus enteritis between January 2010 and December 2021. During the study period, a total of 34,487 infants younger than 12 months were diagnosed with rotavirus enteritis in South Korea. During the two-year COVID-19 pandemic (2020–2021), the rate of decline was significant (5843 cases in 2010 and 1125 in 2019), and by 2021, the total number of patients was almost negligible, as there are only 18 cases in 2021. A significant increase in the ratio of low birth weight (LBW) infants of inpatient department was observed from 2010 to 2021 (4.86% in 2010; 7.77% in 2019; and 23.08% in 2021), indicating that LBW infants are more vulnerable than infants born with normal weight. Average medical expenses related to rotavirus infections also declined significantly from 3,789,443,998 per year (pre-pandemic) to 808,353,795 per year (pandemic). Overall, personal hygiene and social distancing may play important roles in reducing rotavirus infections. However, further studies are needed to determine whether this decreasing trend persists after quarantine and whether the social life of individuals resumes.

## 1. Introduction

Rotavirus is one of the common viruses that can cause acute gastroenteritis in children, especially in those younger than five years [1,2,3]. Rotaviruses are non-enveloped double-stranded RNA viruses that contain 11 segments and are classified into seven serogroups (A to G) and two sub-groups (I or II) [4]. These viruses are further classified based on the antigenicity of the inner capsid protein (VP6). Group A rotaviruses are the most common causes of infections in children and are further subdivided into several genotypes based on the outer capsid glycoprotein VP7 (G-type) and VP4 (P-type) [1,2,3,4]. Although vaccination has decreased the prevalence of Group A rotavirus infections in South Korea, since 2000, Group A strains with the G4P and G8P genotypes have been predominant in children younger than 12 months in group outbreaks [1,2,3,4].

Rotavirus can cause gastroenteritis in children, especially in those younger than five years [1]. Rotavirus infection has various clinical manifestations in children, and patients may be asymptomatic before they exhibit systemic and gastrointestinal symptoms, such as fever, vomiting, diarrhea, and abdominal pain [2]. Patients with severe dehydration may require hospitalization, and in rare but severe cases, may die. Although rotavirus vaccination has been implemented worldwide since 2006, approximately 200,000 people, primarily in low-income countries, die yearly owing to rotavirus infection [3]. Rotavirus can be transmitted through contaminated water or droplets but is commonly transmitted through the fecal–oral route and has an incubation period of <48 h (1–7 days) [4]. Group outbreaks tend to mainly occur in daycare centers, hospitals, and other group settings. Thus, personal hygiene is critical to the prevention of these infections [5].

Owing to the worldwide health crisis attributable to the coronavirus disease (COVID-19) pandemic, which began in December 2019, individual lifestyles have undergone several changes, with emphases on social distancing and personal hygiene [6,7]. The prevalence of upper respiratory tract infections and other gastrointestinal infectious diseases (i.e., norovirus and salmonellosis) decreased in 2020 owing to better hand hygiene practices, wearing of masks, and social distancing [8,9,10]. Thus, personal hygiene may be one of the major factors contributing to the reduced prevalence of various infectious diseases, which indicates that the prevalence of rotavirus enteritis may continue to decrease.

This study aimed to assess the prevalence and characteristics of rotavirus enteritis during the COVID-19 pandemic compared to the 10-year period (2010–2019) before the pandemic. Furthermore, the impact of personal hygiene on public health and the likelihood that personal hygiene could continue to be a significant factor, especially for perinatal health, even after the end of the COVID-19 pandemic were determined.

## 2. Materials and Methods

This nationwide representative study was conducted using data acquired from the National Health Insurance Service (NHIS) from 2010 to 2021 [11,12]. The NHIS provides claims data from the two main health insurance programs, the Korean National Health Insurance system and the Medical Aid program, which cover almost all Korean residents living in South Korea [13]. We identified 34,487 patients younger than 12 months who were diagnosed with rotavirus enteritis between January 2010 and December 2021. Thereafter, we compared the trend in rotavirus prevalence before the COVID-19 pandemic (2010–2019) with that during the same (2020–2021).

The following data were collected from the NHIS database for patients with rotavirus enteritis: sex, age, medications, region of residence (urban vs. rural) [14], household income, use of postpartum care, hospitalization (including intensive care unit), birth weight (normal vs. low birth weight [LBW]), overall medical costs, and individual medical costs.

This study was approved by the NHIS (no. NHIS-2022-1-516) and the Institutional Review Board of Kyung Hee Medical Center; the need for informed consent was waived (IRB no. KHUH 2021-08-010). All claim data provided by the National Health Insurance Sharing Service were de-identified.

The main outcome was rotavirus infection resulting in enteritis, defined as A08.0 according to the International Classification of Diseases, 10th edition.

The following covariates were included in the study: sex, region of residence (urban vs. rural), household income (high, middle, and low), birth weight (normal and LBW), and outpatient and inpatient settings. Household income was divided into three groups: high, middle, and low. The highest 20% (80–100 percentile) were categorized in the high group. People in the 40–79 percentile were categorized in the middle group. The others were categorized in the low group [11]. Birth weight was divided into two groups: children whose birth weight was 2999 g or less were categorized in the LBW group, and the rest of the children were categorized in the normal group. Birth weight was obtained from claim codes.

The data were collected from 2010 to 2021 by the NHIS. The long-term trend change in the number of patients diagnosed with rotavirus enteritis was analyzed [15,16]. The following graphical analysis method was used to determine any difference in the prevalence change between the 10-year period before and during the COVID-19 pandemic (2020–2021). Multiple variables were presented as a percentage of the total count, categorized by year. Data were analyzed by using generalized linear model. The estimation of the beta-estimate was performed. The slope (beta-estimate), which corresponds to the rate of change, was estimated using a generalized linear model from 2010 to 2020. The dependent variables were the outcome variables (such as the total cost), and the independent variable was the year (continuous value). An interaction analysis was performed to identify any difference before and during the pandemic. All statistical analyses were performed using the SAS version 9.4 (SAS Institute Inc., Cary, NC, USA). Significance was set at *p* < 0.05 [17,18].

## 3. Results

During the study period, a total of 34,487 infants younger than 12 months were diagnosed with rotavirus enteritis in South Korea. Most patients were males (54.76%, n = 18,885). Of the patient cohort, 10,289 (29.8%) were outpatients and 24,198 (70.2%) required hospitalization. In terms of household income, 13,203 (38.3%) patients belonged to the low-income class, 14,256 (41.3%) belonged to the middle-income class, and 7028 (20.4%) belonged to the high-income class. The number of patients gradually decreased in each category (Table 1). During the 10-year period prior to the COVID-19 pandemic (January 2010 to December 2019), the total number of infected patients was 33,682. From January 2020 to December 2020, only 787 patients were infected. Remarkably, in 2021, only 18 patients were diagnosed with rotavirus enteritis.

An interaction analysis was conducted to compare the change in the number of infected patients in the past two years with that in the previous 10-year period. The total number of infected patients continuously decreased, with a more rapid decrease in the rate from January 2020 to December 2021 relative to that recorded during the COVID-19 pandemic. This trend may be due to a decline in the overall birth rate; however, the data suggest a more rapid decline since the beginning of the COVID-19 pandemic (Table 1).

Table 2 and Table 3 present the changes in the prevalence of outpatient and inpatient department cases of rotavirus infections, respectively. Among the patients who required hospitalization, the number from urban areas significantly decreased. A decline in the number of patients from rural areas was also observed; however, the decline was not considered significant (urban (for 1000 births) 2010–2019 vs. 2020–2021; *p*-value = 0.0135, rural (for 1000 births) 2010–2019 vs. 2020–2021; *p*-value = 0.0922). Although the total number of male patients (n = 18,885) was greater than the total number of female patients in this study, this difference was only considered significant in the inpatient population (Figure 1).

The medical costs due to rotavirus enteritis were similar from 2010 to 2013 but increased slightly in 2014. As the prevalence of the infection decreased, the total and individual medical costs sharply decreased (sum of individual medical costs in 2010–2019 vs. 2020–2021; *p*-value = 0.0010) (Table 4).

## 4. Discussion

This study sought to assess the 12-year trend change in rotavirus infections in South Korea using the data of patients younger than 12 months who were diagnosed with rotavirus enteritis between 2010 and 2021. The incidence of rotavirus infections indicated a continuous reduction throughout the study period, which aligns with the global trend [3,19,20]. However, during the two-year period of the COVID-19 pandemic (2020–2021), the rate of decline was significant, and by 2021, the total number of patients was almost negligible. A significant increase in the proportion of LBW infants was observed between 2010 and 2021, indicating that LBW infants are more vulnerable than infants born having normal weight. Medical expenses related to rotavirus infections also decreased significantly, especially during the pandemic.

Previous studies reported a constant decrease in rotavirus infections prior to the COVID-19 pandemic. In particular, a study that sought to elucidate the relationship between rotavirus vaccination and rotavirus diarrhea among children in the US suggested a decrease in rotavirus-associated mortality [21]. This study emphasized the importance of rotavirus vaccination in reducing global rotavirus infection. A previous study conducted in South Korea analyzed the changes in rotavirus vaccine coverage and related factors [22] but did not include long-term data and did not discuss the results of increased rotavirus vaccination. Another study performed in South Korea analyzed the impact of rotavirus vaccination in Korea from 2008 to 2020 [23]. Although this study used long-term data, the impact of the COVID-19 pandemic was not assessed.

Similar to other infectious diseases, quarantine and social distancing are important factors contributing to this phenomenon [24]. The high national rotavirus vaccination rate [23] may have contributed to the decreasing number of patients. Data from Statistics Korea indicate that the already low national birth rate began to decline at an accelerated rate in 2000, and the total number of live births has continuously declined from 470,171 in 2010 to 260,562 in 2021 [25].

Owing to the nature of rotaviruses, young children are mainly infected with these viruses [26]. Notably, the declining birth rate in South Korea [27] may have contributed to the decrease in the number of infected patients. However, due to the impact of quarantine during the COVID-19 pandemic, the decrease in rotavirus infections may be steeper than the total live births. Most mothers and newborns in Korea use postpartum care centers after birth, and the duration of their stay varies from 2 to 4 weeks [28]. Previously, rotavirus infections often occurred in postpartum care centers. In addition to rotavirus infections, other infectious diseases, such as respiratory syncytial virus infections, have occurred due to group acceptance of newborns. However, during the COVID-19 period, postpartum care centers focused more on quarantine and infection prevention. As a result, outsiders and newborn fathers were sometimes denied access to the hospital. These efforts may have reduced the incidence of rotavirus enteritis.

Although the incidence of rotavirus enteritis decreased among patients younger than 12 months, the ratio of LBW to total rotavirus enteritis cases increased significantly. This result may be attributed to the high vulnerability of LBW babies to general child health problems and disease [29]. The increased vaccination rate during the COVID-19 pandemic might not have affected LBW babies as much as normal babies.

As the infection rate decreased owing to the aforementioned factors, medical expenses also decreased significantly, primarily due to a decrease in the total number of patients. However, the limitations and difficulty of hospitalization during the pandemic may have also contributed to the decrease in costs.

The results of this study imply that the quarantine measures established due to the COVID-19 pandemic have positively influenced the decrease in rotavirus enteritis cases. However, policies that inform citizens regarding the importance of hygiene to child health are still needed. The government should also consider the greater vulnerability of LBW infants to infectious diseases, such as rotavirus enteritis. Notably, the prevalence of infectious diseases has recently increased (i.e., after the quarantine period). Thus, a study on the prevalence of rotavirus enteritis after quarantine is warranted. By performing a prospective study to determine whether rotavirus infections will continue to decrease or increase, an efficient method to reduce the social costs caused by rotavirus enteritis can be identified.

This study was a nationwide, long-term study that used credible data from the NHIS. However, this study had some limitations. First, only patients younger than 12 months who were diagnosed with rotavirus enteritis were included. However, this group of newborns does not represent the general population. Thus, the infection rate among all newborns could not be directly deduced in this study, which makes it difficult to accurately compare the infection rate in this cohort to that of the general population. Nevertheless, as the national medical insurance covers almost the entire population, an indirect comparison was made using the total number of births provided by the NHIS; the prevalence of rotavirus infection was confirmed to decrease more than the birth rate. Second, several rotavirus infections may have been undetermined as these cases may have gone unreported, owing to difficulties in accessing medical facilities during the COVID-19 period. Additional requirements for COVID-19 testing and hospitalization restrictions hindered access to medical facilities. Thus, the decrease in infection rate may be further exaggerated. Furthermore, not all individuals adhered to the prescribed protocols for proper hand hygiene and maintaining appropriate social distances. Third, the data we utilized is close to the general population, since NHIS includes 98% of the entire Korean population. However, people who may be more vulnerable to rotaviral infection than the general population, such as immigrants or illegal immigrants, may be left out of the data [11]. Finally, rotavirus-related vaccination rates could not be measured in our nationwide cohort. Rotavirus vaccination was first implemented in Korea in 2007. Before 6 March 2023, rotavirus vaccination was not included in the National Immunization Program. However, voluntary vaccination was offered, resulting in a high vaccination rate that is similar to that of other developed countries, such as the US, England, and Australia [22]. Accordingly, the prevalence of rotavirus infections has decreased [30]. Overall, the main results of this study must be carefully interpreted.

## 5. Conclusions

The incidence of rotavirus enteritis was found to decrease over the data collection years. Of note, the COVID-19 pandemic positively influenced the decrease in rotavirus enteritis cases. Based on our results, LBW infants were less affected by the quarantine measures imposed during the COVID-19 pandemic. Moreover, the overall expenses for rotavirus infections decreased over the evaluation period. Therefore, further research on rotavirus infections after quarantine is needed. In addition, political solutions are required to maintain the rotavirus enteritis rate at the expense of treatment.

## Figures and Tables

**Figure 1 children-10-01436-f001:**
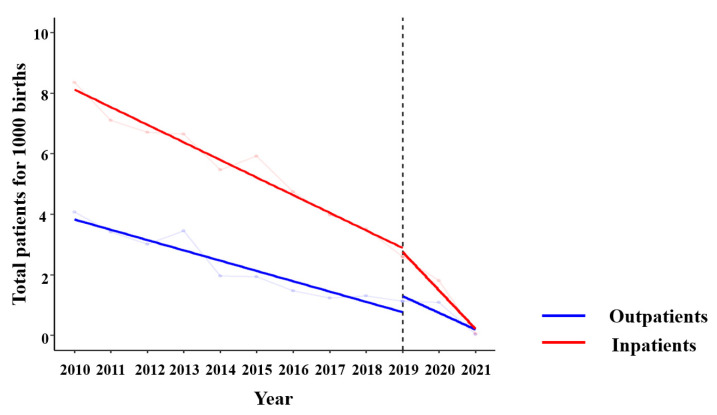
Prevalence and rate of change of the total outpatient and inpatient cases (2010–2019 vs. 2020–2021). The *y*-axis (slope) corresponds to the rate of change.

**Table 1 children-10-01436-t001:** Baseline characteristics of the study participants.

Variables	2010	2011	2012	2013	2014	2015	2016	2017	2018	2019	2020	2021
Total patients	5843	4955	4711	4410	3238	3445	2521	1861	1573	1125	787	18
Sex (n, %)												
Male	3178 (54.39)	2751 (55.52)	2602 (55.23)	2369 (53.72)	1782 (55.03)	1899 (55.12)	1386 (54.98)	1005 (54.00)	829 (52.70)	649 (57.69)	427 (54.26)	08 (44.44)
Female	2665 (45.61)	2204 (44.48)	2109 (44.77)	2041 (46.28)	1456 (44.97)	1546 (44.88)	1135 (45.02)	856 (46.00)	744 (47.30)	476 (42.31)	360 (45.74)	10 (55.56)
Region of residence (n, %)												
Urban	2507 (42.91)	2332 (47.06)	2160 (45.85)	2181 (49.46)	1428 (44.10)	1455 (42.24)	1144 (45.38)	762 (40.95)	678 (43.10)	492 (43.73)	321 (40.79)	9 (50.00)
Rural	3336 (57.09)	2623 (52.94)	2551 (54.15)	2229 (50.54)	1810 (55.90)	1990 (57.76)	1377 (54.62)	1099 (59.05)	895 (56.90)	633 (56.27)	466 (59.21)	9 (50.00)
Household income (n, %)												
Low	2410 (41.25)	2027 (40.91)	1820 (38.63)	1601 (36.30)	1249 (38.57)	1254 (36.40)	914 (36.26)	671 (36.06)	559 (35.54)	413 (36.71)	279 (35.45)	6 (33.33)
Middle	2383 (40.78)	1973 (39.82)	1998 (42.41)	1883 (42.70)	1317 (40.67)	1460 (42.38)	1058 (41.97)	761 (40.89)	643 (40.88)	456 (40.53)	318 (40.41)	6 (33.33)
High	1050 (17.97)	955 (19.27)	893 (18.96)	926 (21.00)	672 (20.75)	731 (21.22)	549 (21.78)	429 (23.05)	371 (23.59)	256 (22.76)	190 (24.14)	6 (33.33)
Patient department (n, %)												
OPD	1913 (32.74)	1607 (32.43)	1457 (30.93)	1506 (34.15)	855 (26.41)	850 (24.67)	596 (23.64)	440 (23.64)	424 (26.95)	340 (30.22)	296 (37.61)	5 (27.78)
IPD	3930 (67.26)	3348 (67.57)	3254 (69.07)	2904 (65.85)	2383 (73.59)	2595 (75.33)	1925 (76.36)	1421 (76.36)	1149 (73.05)	785 (69.78)	491 (62.39)	13 (72.22)

IPD, inpatient department; NHISS, National Health Insurance Sharing Service; OPD, outpatient department.

**Table 2 children-10-01436-t002:** Prevalence of outpatient department (OPD) cases of rotavirus enteritis, 2010–2021 (n = 10,289).

Variables (n, %)	2010	2011	2012	2013	2014	2015	2016	2017	2018	2019	2020	2021
Total OPD	1913	1607	1457	1506	855	850	596	440	424	340	296	5
Sex												
Male	1018 (53.21)	901 (56.07)	767 (52.64)	813 (53.98)	444 (51.93)	480 (56.47)	320 (53.69)	239 (54.32)	217 (51.18)	196 (57.65)	150 (50.68)	3 (60.00)
Female	895 (46.79)	706 (43.93)	690 (47.36)	693 (46.02)	411 (48.07)	370 (43.53)	276 (46.31)	201 (45.68)	207 (48.82)	144 (42.35)	146 (49.32)	2 (40.00)
Region of residence												
Urban	713 (37.27)	678 (42.19)	639 (43.86)	811 (53.85)	376 (43.98)	327 (38.47)	276 (46.31)	171 (38.86)	166 (39.15)	120 (35.29)	114 (38.51)	1 (20.00)
Rural	1200 (62.73)	929 (57.81)	818 (56.14)	695 (46.15)	479 (56.02)	523 (61.53)	320 (53.69)	269 (61.14)	258 (60.85)	220 (64.71)	182 (61.49)	4 (80.00)
Household income												
Low	758 (39.62)	647 (40.26)	519 (35.62)	493 (32.74)	313 (36.61)	301 (35.41)	218 (36.58)	150 (34.09)	149 (35.14)	118 (34.71)	102 (34.46)	1 (20.00)
Middle	801 (41.87)	629 (39.14)	657 (45.09)	667 (44.29)	344 (40.23)	358 (42.12)	241 (40.44)	183 (41.59)	174 (41.04)	147 (43.24)	117 (39.53)	2 (40.00)
High	354 (18.50)	331 (20.60)	281 (19.29)	346 (22.97)	198 (23.16)	191 (22.47)	137 (22.99)	107 (24.32)	101 (23.82)	75 (22.06)	77 (26.01)	2 (40.00)
Birth weight												
Normal	1770 (92.52)	1461 (90.91)	1311 (89.98)	1411 (93.69)	753 (88.07)	740 (87.06)	503 (84.40)	388 (88.18)	376 (88.68)	308 (90.59)	258 (87.16)	5 (100.00)
LBW	143 (7.48)	146 (9.09)	146 (10.02)	95 (6.31)	102 (11.93)	110 (12.94)	93 (15.60)	52 (11.82)	48 (11.32)	32 (9.41)	38 (12.84)	0 (0.00)

LBW, low birth weight; NHISS, National Health Insurance Sharing Service; OPD, outpatient department.

**Table 3 children-10-01436-t003:** Prevalence of inpatient department (IPD) cases of rotavirus enteritis, 2010–2021 (n = 24,198).

Variables (n, %)	2010	2011	2012	2013	2014	2015	2016	2017	2018	2019	2020	2021
Total IPD	3930	3348	3254	2904	2383	2595	1925	1421	1149	785	491	13
Sex												
Male	2160 (54.96)	1850 (55.26)	1835 (56.39)	1556 (53.58)	1338 (56.15)	1419 (54.68)	1066 (55.38)	766 (53.91)	612 (53.26)	453 (57.71)	277 (56.42)	5 (38.46)
Female	1770 (45.04)	1498 (44.74)	1419 (43.61)	1348 (46.42)	1045 (43.85)	1176 (45.32)	859 (44.62)	655 (46.09)	537 (46.74)	332 (42.29)	214 (43.58)	8 (61.54)
Region of residence												
Urban	1794 (45.65)	1654 (49.40)	1521 (46.74)	1370 (47.18)	1052 (44.15)	1128 (43.47)	868 (45.09)	591 (41.59)	512 (44.56)	372 (47.39)	207 (42.16)	8 (61.54)
Rural	2136 (54.35)	1694 (50.60)	1733 (53.26)	1534 (52.82)	1331 (55.85)	1467 (56.53)	1057 (54.91)	830 (58.41)	637 (55.44)	413 (52.61)	284 (57.84)	5 (38.46)
Household income												
Low	1652 (42.04)	1380 (41.22)	1301 (39.98)	1108 (38.15)	936 (39.28)	953 (36.72)	696 (36.16)	521 (36.66)	410 (35.68)	295 (37.58)	177 (36.05)	5 (38.46)
Middle	1582 (40.25)	1344 (40.14)	1341 (41.21)	1216 (41.87)	973 (40.83)	1102 (42.47)	817 (42.44)	578 (40.68)	469 (40.82)	309 (39.36)	201 (40.94)	4 (30.77)
High	696 (17.71)	624 (18.64)	612 (18.81)	580 (19.97)	474 (19.89)	540 (20.81)	412 (21.40)	322 (22.66)	270 (23.50)	181 (23.06)	113 (23.01)	4 (30.77)
Birth weight												
Normal	3739 (95.14)	3129 (93.46)	2992 (91.95)	2698 (92.91)	2143 (89.93)	2362 (91.02)	1762 (91.53)	1294 (91.06)	1035 (90.08)	724 (92.23)	449 (91.45)	10 (76.92)
LBW	191 (4.86)	219 (6.54)	262 (8.05)	206 (7.09)	240 (10.07)	233 (8.98)	163 (8.47)	127 (8.94)	114 (9.92)	61 (7.77)	42 (8.55)	3 (23.08)

IPD, inpatient department; NHISS, National Health Insurance Sharing Service; LBW, low birth weight.

**Table 4 children-10-01436-t004:** Total patients, OPD, IPD, and total expense due to rotavirus enteritis pre-COVID-19 and during the COVID-19 pandemic, 2010–2021 (n = 34,487).

	Pre-COVID-19 Pandemic	During the COVID-19 Pandemic	Total
Total patients, n	33,682	805	34,487
Average patients per year, n	3682	403	2874
Total OPD, n	9988	301	10,289
Total IPD, n	23,694	504	24,198
Total expense, KRW	37,894,439,980	1,616,707,590	39,511,147,570
Average expense per year, KRW	3,789,443,998	808,353,795	3,292,595,630

OPD, outpatient department; IPD, inpatient department. 1000 KRW is currently about $0.76.

## Data Availability

Data are available upon reasonable request. Study protocol, statistical code, and data set: available from DKYon (email: yonkkang@gmail.com).

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
