# Peer review of "National Trends in Rotavirus Enteritis among Infants in South Korea, 2010–2021: A Nationwide Cohort"

_children, 2023, doi:10.3390/children10091436_

Round 1

Reviewer 1 Report

The paper is pretty short, but well written and Ad Hoc with the aim and scope of the journal. Authors analyze data on how rotavirus infection in children from South Korea decreased during pandemic period with proper statistics. This short report may be useful for deeper studies in the same topic and may be a reference for studies performed in other countries.  

I only suggest to address the following minor concerns:

1. English should be brushed up. Several style errors are detected at first glance 

2. Presentation of Figures and references must be revised since these are not prepared according to journal style. Footnotes are not correctly located. In addition axis of figure 1 do not contain any descriptor. Please add it and improve Figure resolution (please increase font size). 

Minor editing

Reviewer 2 Report

This study, similar to other observations made for other non-SARS-CoV-2 viral infections during the COVID-19 pandemic demonstrated in a large South Korean sample a reduction in rotavirus cases in subjects less than one year old with a collateral also cost savings associated with such infections.

The study, overall, is well presented but there are some elements to consider:

INTRODUCTION

- The introduction is too meager: it is useful to provide a microbiological definition of rotavirus. Express which strains are most active and widespread in South Korea. Provide a brief but more detailed digression on the routes of transmission of this virus;

- It would be useful to search for studies that showed a reduction during the pandemic from COVID-19 in the incidence of additional gastrointestinal infections (appart from rotavirus) to express that this trend was also formed for other infections.

- At the end of the introduction you say that you want to assess how the impact of hygiene measures resulted on rotavirus infection. That seems to me a tad to be downplayed. it is clear that in the pandemic period such measures were taken. But we are talking about people aged < 12 months. How can you guarantee that in the entire sample everyone followed these measures? At the very least, it should be discussed.

MATERIALS AND METHODS

- I think the statistical analysis section should be more detailed;

- Did you have data misses on the entire sample? If so, how did you handle them?

RESULTS

- You define "low-income class, middle-income" but no monetary thresholds are defined in the methods. I recommend that in the methods this is laid out well on how the variables were weighted;

- Did you compare subgroups based on income? Both before and after the pandemic? Certainly these are exploratory analyses being subgroup comparisons but nevertheless you have a large numerosity so why not do them?

- The same thing can be done based on the other main variables.

- What monetary units did you use in Table 4?

- Figure 1 needs to have a much more detailed legend.  The reader needs to understand how the figure should be read. Can data on statistical significance also be included on that figure? "Ex. * < 0.05 and the like?"

Please review the manuscript linguistically and grammatically.

Round 2

Reviewer 2 Report

Thank to authors for having made the appropriate revisions.

Author Response

Thank you for your criticial comments. I look forward to working with you and the reviewers to move this manuscript closer to publication in the Children.